# Combined Transfibular and Anterior Approaches Increase Union Rate and Decrease Non-Weight-Bearing Periods in Ankle Arthrodesis: Combined Approaches in Ankle Arthrodesis

**DOI:** 10.3390/jcm10245915

**Published:** 2021-12-16

**Authors:** Jeong-Jin Park, Whee-Sung Son, In-Ha Woo, Chul-Hyun Park

**Affiliations:** 1Department of Orthopaedic Surgery, Yeungnam University Medical Center, Daegu 42415, Korea; wjdwls3912@naver.com (J.-J.P.); buonggiorno39@gmail.com (I.-H.W.); 2Department of Orthopaedic Surgery, Guro Hospital, Korea University Medical Center, Seoul 08308, Korea; oswsson@gmail.com; 3Department of Orthopaedic Surgery, College of Medicine, Yeungnam University, Daegu 38541, Korea

**Keywords:** ankle, end-stage arthritis, arthrodesis, approach

## Abstract

The transfibular approach is a widely used method in ankle arthrodesis. However, it is difficult to correct coronal plane deformity. Moreover, it carries a risk of nonunion and requires long periods of non-weight-bearing because of its relatively weak stability. We hypothesized that the transfibular approach combined with the anterior approach in ankle arthrodesis wound yield a higher fusion rate and shorter non-weight-bearing period. This study was performed to evaluate the clinical and radiographic results and postoperative complications in ankle arthrodesis using combined transfibular and anterior approaches in end-stage ankle arthritis. Thirty-five patients (36 ankles) with end-stage ankle arthritis were consecutively treated using ankle arthrodesis by combined transfibular and anterior approaches. The subjects were 15 men and 20 women, with a mean age of 66.5 years (46–87). Clinical results were assessed using the visual analog scale (VAS) for pain, the American Orthopaedic Foot and Ankle Society (AOFAS) scores, and the ankle osteoarthritis scale (AOS) preoperatively and at the last follow-up. Radiographic results were assessed with various radiographic parameters on ankle weight-bearing radiographs and hindfoot alignment radiographs. All clinical scores significantly improved after surgery. Union was obtained in all cases without additional surgery. Talus center migration (*p* = 0.001), sagittal talar migration (*p* < 0.001), and hindfoot alignment angle (*p* = 0.001) significantly improved after surgery. One partial skin necrosis, two screw penetrations of the talonavicular joint, and four anterior impingements because of the bulky anterior plate occurred after surgery. In conclusion, combined transfibular and anterior approaches could be a good method to increase the union rate and decrease the non-weight-bearing periods in ankle arthrodesis.

## 1. Introduction

Ankle arthrodesis has been commonly used in the treatment of end-stage ankle arthritis because of its straightforward procedure and suitability for almost all types of ankle arthritis. Moreover, there is high level of evidence in the literature for the use of ankle arthrodesis in the surgical treatment of end-stage ankle arthritis [1,2,3]. Several approaches have been commonly used for ankle arthrodesis, including the anterior, transfibular, posterior, and arthroscopic approaches [4,5,6].

The transfibular approach has been widely used in recent years because it provides a wide surgical field, and bone grafting can be performed using the resected fibula [5]. However, it is difficult to correct the coronal plane deformity and debride the medial gutter using the transfibular approach [7]. In ankle arthrodesis using the transfibular approach, the arthrodesis site is commonly fixed using transarticular screws. However, screw fixation carries a risk of nonunion and requires long periods of non-weight-bearing after surgery because of its relatively weak stability. In a recent systematic review, the average union rate following ankle arthrodesis was 89% (range: 64–100%) [8]. In addition, it is commonly recommended in the literature to keep non-weight-bearing for 8 to 12 weeks after surgery [6,7]. These are related to the rigidity of the arthrodesis construct. Therefore, various methods have been attempted to increase the stability of the arthrodesis site [9,10,11,12].

In biomechanical studies, anterior plate supplementation increases construct rigidity and decreases micromotion [10]; however, clinical studies are sparse. The authors recently performed ankle arthrodesis using a transfibular approach and additional anterior plating using an anterior approach, and they allowed full weight-bearing at 4 weeks after surgery. These combined approaches have the advantages of allowing debridement of the medial gutter, correcting coronal plane deformity, and adding anterior plating through an anterior incision. To the best of our knowledge, no study has yet reported on the effectiveness of combined transfibular and anterior approaches.

We hypothesized that the transfibular approach combined with the anterior approach in ankle arthrodesis would yield a higher fusion rate and shorter non-weight-bearing period compared with previous reports about the transfibular approach alone. The purpose of this study was to evaluate the clinical and radiographic results and postoperative complications of ankle arthrodesis using combined transfibular and anterior approaches in end-stage ankle arthritis.

## 2. Materials and Methods

This study was approved by the Institutional Review Board of Yeungnam University Hospital, and informed consent was waived because of its retrospective design. Between February 2012 and November 2017, 35 patients (36 ankles) with end-stage ankle arthritis of modified Takakura stage 4 were consecutively treated using ankle arthrodesis by a single surgeon. All surgeries were performed using combined transfibular and anterior approaches. All patients had persistent ankle pain with at least 3 months of failed nonoperative treatments, including immobilization, medications, and ankle braces. Patients with pyogenic arthritis, failed ankle arthroplasty, and neuropathic arthritis were excluded. The follow-up period was 25.6 months (13–60). The subjects were 15 men and 20 women, with a mean age of 66.5 years (46–87) and a mean body mass index of 25 (16.4–31.2). Six patients had diabetes and one was a smoker. All cases, except three post-traumatic arthritis cases, were of degenerative arthritis.

Clinical results were assessed using the visual analog scale (VAS) for pain, the American Orthopedic Foot and Ankle Society (AOFAS) ankle-hindfoot functional score, and the ankle osteoarthritis scale (AOS) preoperatively and at the last follow-up. Operation time and postoperative complications were evaluated by chart review.

Radiograph results were evaluated using weight-bearing ankle anteroposterior and lateral radiographs and hindfoot alignment radiographs preoperatively and at the last follow-up. All radiographs were obtained digitally, and the radiographic parameters were measured using the Picture Archiving Communication System (Infinity, Seoul, Korea), including medial distal tibial angle (MDTA), talar tilt, talar center migration, anterior distal tibial angle (ADTA), anteroposterior offset, sagittal talar migration, and hindfoot alignment angle (Figure 1). MDTA and ADTA were defined as the angles between the tibial axis and the tibial plafond on anteroposterior and lateral radiographs. The tibial axis was drawn by connecting the midpoint between the cortex at 5 and 10 cm proximal to the joint line on anteroposterior and lateral radiographs. The talar tilt was defined as the angle between the tibial plafond and the talar dome on anteroposterior radiographs, and a positive number indicated a varus tilt of the talus. Talus center migration was defined as the shortest distance between the tibial axis and the center of the talus on ankle anteroposterior radiograph, and a positive number implied that the talus center was located medial to the tibial axis [13]. Anteroposterior offset was used to assess the anterior translation of talus relative to the axis of the tibia in the sagittal plane [14]. Anteroposterior offset was defined as the distance between the tibial axis and the center of the talus on lateral radiograph and was evaluated only preoperatively, because it was difficult to accurately determine the center of the talus in the postoperative radiographs taken after union. In order to assess the correction of the anterior translation of the talus in the sagittal plane after surgery, sagittal talar migration was used. Sagittal talar migration was measured as the distance between the longitudinal axis of the tibia and the lateral process of the talus on lateral radiographs [15]. Anteroposterior offset and sagittal talar migration were represented as a positive number when the center or the lateral process of the talus was located anterior to the tibial axis. The hindfoot alignment angle was defined as the angle between the tibial and the calcaneal axis on hindfoot alignment radiograph. The calcaneal axis was defined as the bisecting line of the angle formed by two lines representing the lateral and medial osseous contours of the calcaneus [16]. A positive number meant that the hindfoot was varus oriented.

Surgery was performed with the patient in the lateral decubitus position, and it was possible to change to the supine position during the anterior approach. All patients received prophylactic antibiotics before surgery. An 8 to 10 cm lateral incision was made along the posterior border of the lateral malleolus to obtain a sufficient distance from the anterior incision (Figure 2). Soft tissue dissection was performed between the periosteum and the distal fibula to preserve the blood supply of the soft tissue. After exposure of the distal fibula, the fibula was transected 4 to 5 cm proximal to the ankle joint (Figure 3). The cartilage and subchondral bone of the tibiotalar joint were removed using curettes, osteotomes, and burrs through the lateral ankle joint (Figure 4). Following joint preparation, an anterior longitudinal incision of 6 to 8 cm was made by keeping a distance of 7 cm from the lateral incision. After dissection was performed between the tibialis anterior and extensor hallucis longus tendons to expose the joint, cartilage and subchondral bone of the medial gutter were removed (Figure 5). While the tibiotalar joint was being prepared, the resected fibular fragment was hemisected in the sagittal plane for use as a bone graft (Figure 6). The medial half was used as the filler of the arthrodesis site, and the lateral half was used as a lateral strut. The tibiotalar joint was reduced in a neutral and plantigrade position and temporarily fixed using a 2.4 mm-Steinmann pin. Subsequently, the arthrodesis site was fixed using the first 6.5-mm partially threaded cannulated screw from the posterior malleolus along the longitudinal axis of the talar neck and into the talar head as a homerun screw. The second 6.5-mm cannulated screw was placed from the metaphyseal flare of the medial distal tibia into the medial talar body. After screw fixation, additional anterior plating was performed using a 3.5-mm T-plate (Synthes, Paoli, PA, USA) in 29 ankles and an anatomical anterior fusion plate (Arthrex Inc., Naples, FL, USA) in 7 ankles through anterior incision (Figure 7). The lateral half of the resected fibula was fixed by four cancellous screws in its original anatomical position as a natural lateral bony plate (Figure 8). Reduction state and screw positioning without penetration of the subtalar joints were confirmed via fluoroscopy (Figure 9). The extensor retinaculum was meticulously closed with 2-0 absorbable sutures, followed by subcutaneous closure with 3-0 absorbable suture. The skin was closed with 4-0 monofilament sutures using the modified Allgöwer–Donati technique. Short leg casts were applied for 4 weeks following surgery. At 2 weeks after surgery, partial weight-bearing was started with crutches. Full weight-bearing with and without an ankle orthosis was allowed at 4 weeks and 8 weeks after surgery, respectively.

All dependent variables were tested for normality of distribution, and data normality was determined using a Kolmogorov–Smirnov test. Student’s *t*-test was used for comparison of pre- and postoperative results. For all tests, *p* values < 0.05 were considered significant.

## 3. Results

VAS, AOFAS ankle-hindfoot functional score, and AOS improved from 6.8 (6–9), 56.2 (41–75), and 67.5 (51.1–88.1) preoperatively to 2.6 (0–7), 80.7 (61–92), 25.6 (5–60.6) at the last follow-up, respectively. The mean operation time was 161 min (125–225).

Union was obtained in all cases without additional surgery. The pre- and postoperative radiographic results are shown in Table 1. A total of 15 ankles (41.7%) had talar tilt of more than 15°, and 10 ankles (27.8%) had talar center migration of more than 8 mm in the coronal plane preoperatively (Figure 10). Eight ankles (22.2%) had an anteroposterior offset of more than 10 mm preoperatively. Talus center migration (*p* = 0.001), sagittal talar migration (*p* < 0.001), and hindfoot alignment angle (*p* = 0.001) significantly improved after surgery (Figure 11).

One ankle had partial skin necrosis that healed without additional surgical procedures. Two ankles had pain in the talonavicular joint due to screw penetration after surgery, and the pain decreased after screw removal. Four ankles that were fixed using an anatomical anterior fusion plate had impingement pain because of a bulky anterior plate, and the pain resolved after plate removal.

## 4. Discussion

This study was performed to evaluate the clinical and radiographic results and postoperative complications of ankle arthrodesis using combined transfibular and anterior approaches in end-stage ankle arthritis. In the present study, even though early weight-bearing rehabilitation was performed, union was achieved in all cases.

Ankle arthrodesis and ankle arthroplasty are effective treatment options for end-stage ankle arthritis [1]. However, compared to ankle arthroplasty, ankle arthrodesis has two obvious weaknesses, including the risk of nonunion and a need for longer non-weight-bearing period. When performing joint arthrodesis, it is essential to keep in mind the following principles for reducing the non-weight-bearing period and increasing the union rate: (1) thorough joint preparation, (2) deformity correction with good alignment of the arthrodesis site, (3) stable and rigid fixation, and (4) compression across the arthrodesis site [7].

It is important to choose an appropriate approach for joint preparation and deformity correction when performing ankle arthrodesis. Several surgical approaches, including the anterior, transfibular, posterior, and minimally invasive approaches, have been described in the literature [4,5,6]. The anterior approach allows great exposure to the anterior ankle joint and is easy to correct for coronal plane deformities. This approach has a limitation, in that it lacks exposure to the posterior ankle joint and the malleoli [7]. The minimal invasive or mini-open arthrotomy approach has the advantage of the open and arthroscopic ankle arthrodesis methods while limiting their respective disadvantages. However, this approach has limited utility in cases with significant deformity or bone loss [17,18]. The transfibular approach is a common technique used today because of excellent visualization of the lateral joint and easy correction of sagittal deformities [5,19]. In addition, this approach allows for fixation using a fibular strut graft, which can provide additional stability, and an autogenous bone graft using the resected half of the fibula without harvesting the iliac bone. However, the transfibular approach has some disadvantages, such as difficulties in debriding a medial gutter, correcting for coronal plane deformity, and performing plate fixation. In particular, when the valgus deformity of the plafond is severe, it is difficult to perform a flat cut parallel to the floor with the foot in neutral standing position through the transfibular approach, and there is a risk of inducing iatrogenic fracture of medial malleoli [20]. If coronal plane malunion occurs, there is a risk of secondary arthritis of subtalar arthritis and unacceptable overloading of lateral border of the foot [21]. Therefore, in the present study, an anterior approach was additionally performed for debridement of the medial gutter, correction of coronal deformity, and additional anterior plating.

Nonunion is the most serious complication after joint arthrodesis. Historically, nonunion rates have been as high as 40%, largely because of destabilizing surgical techniques [2]. A recent meta-analysis reported that the average union rate following ankle arthrodesis using modern surgical techniques was 89% (range: 64–100%) [8]. There are several factors that influence joint union, including patients’ comorbidity, thorough preparation of the arthrodesis bed, rigidity of fixation, and patient compliance with postoperative rehabilitation.

Transarticular fixation using two or three large-fragment, partially threaded, cancellous screws is a widely used method in ankle arthrodesis [4,7]. Transarticular fixation has the advantage, in that it can be performed simply, but it has the disadvantage, in that it requires a long non-weight-bearing period of 8–12 weeks after surgery because of its low stability [6,7]. Long-term non-weight-bearing can cause loss of muscle strength and poor quality of life. In addition, especially in elderly patients with poor compliance, there is a high risk of failure of fixation due to early weight-bearing. Therefore, it was introduced to perform additional anterior plating and increase the stability of the arthrodesis site, and its effectiveness has been reported through biomechanical and clinical studies [10,22,23,24]. In this study, we tried to thoroughly debride the cartilage of the medial gutter, which is difficult to reach using a transfibular approach, through an additional anterior approach. After joint preparation, two 6.5-mm partially threaded screws were fixed to the medial and posterior aides, and additional anterior plating was performed through an anterior approach. Finally, the lateral half of the resected fibula was fixed to the lateral side as a bony plate. We believe that sufficient stability could be provided to the arthrodesis site through fixation in four directions. Therefore, we performed early weight-bearing rehabilitation, which allowed partial weight-bearing at 2 weeks after surgery and full weight-bearing at 4 weeks after surgery. Joint union was obtained in all cases without additional surgery, despite early weight-bearing.

Sagittal deformity was readily correctable through the transfibular approach. The tibiotalar joint is inherently stable in the sagittal plane because of its conforming anatomy. However, as arthritis progresses, anterior displacement of the talus often occurs relative to the tibia in the sagittal plane [25,26]. When evaluating sagittal talar migration, it is common to measure the anteroposterior offset, defined as the distance between the tibial axis and the talar head center. However, in these situations, it is difficult to accurately determine the talar head center in the postoperative radiograph taken after bone union. Therefore, in this study, anterior translation of the talus was evaluated using anteroposterior offset preoperatively, and the reduction of the anterior translation of the talus was evaluated using sagittal talar migration. Eight ankles (22.2%) had an anteroposterior offset of more than 10 mm preoperatively, and sagittal talar migration was corrected from 13.6 mm preoperatively to 7.9 mm of neutral mechanical alignment.

It is generally accepted that the coronal plane deformity is difficult to correct through a transfibular approach, since it is difficult to evaluate the degree of correction. In addition, particularly in ankle arthritis with valgus tibial plafond, sawing from the lateral to medial direction carries the risk of creating a fracture in the medial malleolus. In the present study, appropriate tibiotalar alignment could be restored through an anterior approach, although 15 ankles (41.7%) had severe coronal deformity, which was more than 15° of the preoperative coronal deformity. Despite severe incongruence at the tibiotalar joint, we were able to restore appropriate tibiotalar alignment in most cases through an anterior approach. The additional anterior approach allows full access to the ankle joint, allowing complete gutter debridement and ligamentous release as necessary, permitting restoration of mechanical alignment.

Wound problems are one of the most serious complications of our procedure because of the use of two surgical incisions. When using two surgical incisions in the tibial and fibular fractures, several studies highlighted that a 7-cm skin bridge should be preserved between surgical incisions to minimize the risk of wound problems [27,28]. In the present study, a lateral incision was made along the posterior margin of the lateral malleolus to obtain enough distance between the two incisions. Thus, we experienced only one case of partial necrosis that was treatable without additional surgical procedures.

The biggest limitation of this study is the lack of a control group with ankle arthrodesis treated using the transfibular approach alone. However, this study was comparable to previously reported studies with respect to clinical results, union rate, and postoperative complications [19,29,30,31]. The relatively small sample size and retrospective nature are also limitations of this study. Therefore, we believe that a prospective randomized controlled trial with a large sample size is needed to minimize the bias.

## 5. Conclusions

Combined transfibular and anterior approaches showed satisfactory clinical and radiographic results without nonunion or fixation failure. Therefore, this technique could be a good method to increase union rate and decrease non-weight-bearing periods in ankle arthrodesis.

## Figures and Tables

**Figure 1 jcm-10-05915-f001:**
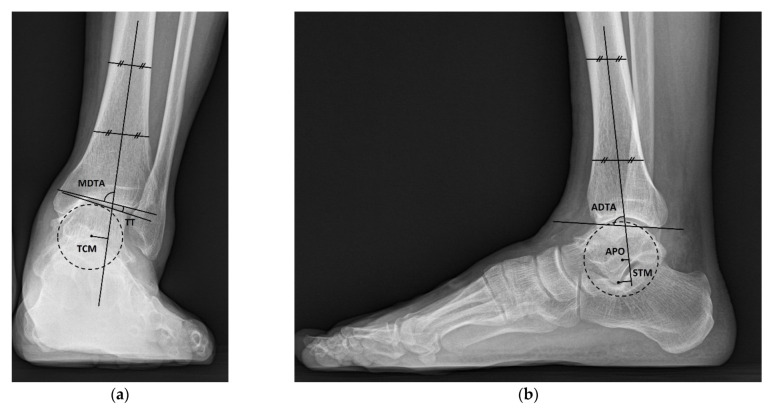
(**a**) Weightbearing ankle anteroposterior radiograph shows measurement of the medial distal tibial ankle (MDTA), the talar tilt (TT), and the talus center migration (TCM). (**b**) Weightbearing foot lateral radiograph shows measurement of the anterior distal tibial angle (ADTA), the anteroposterior offset (APO), and the sagittal talar migration (STM).

**Figure 2 jcm-10-05915-f002:**
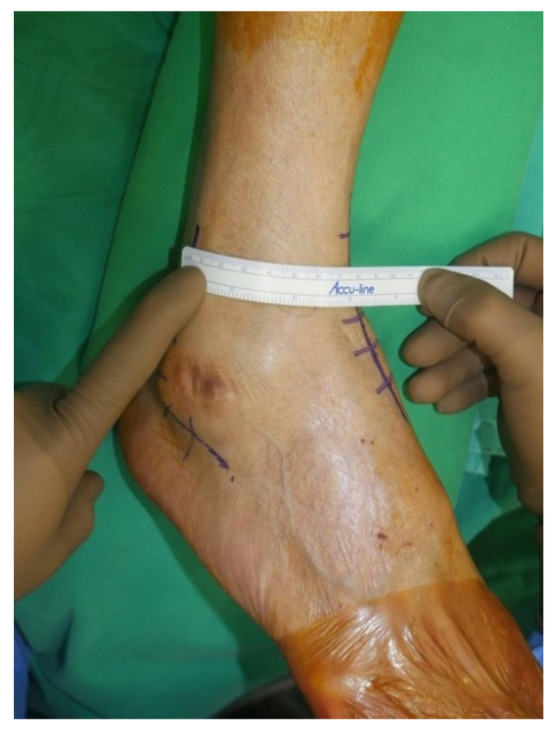
An 8 to 10 cm lateral incision was made along the posterior border of the lateral malleolus to obtain a sufficient distance from the anterior incision.

**Figure 3 jcm-10-05915-f003:**
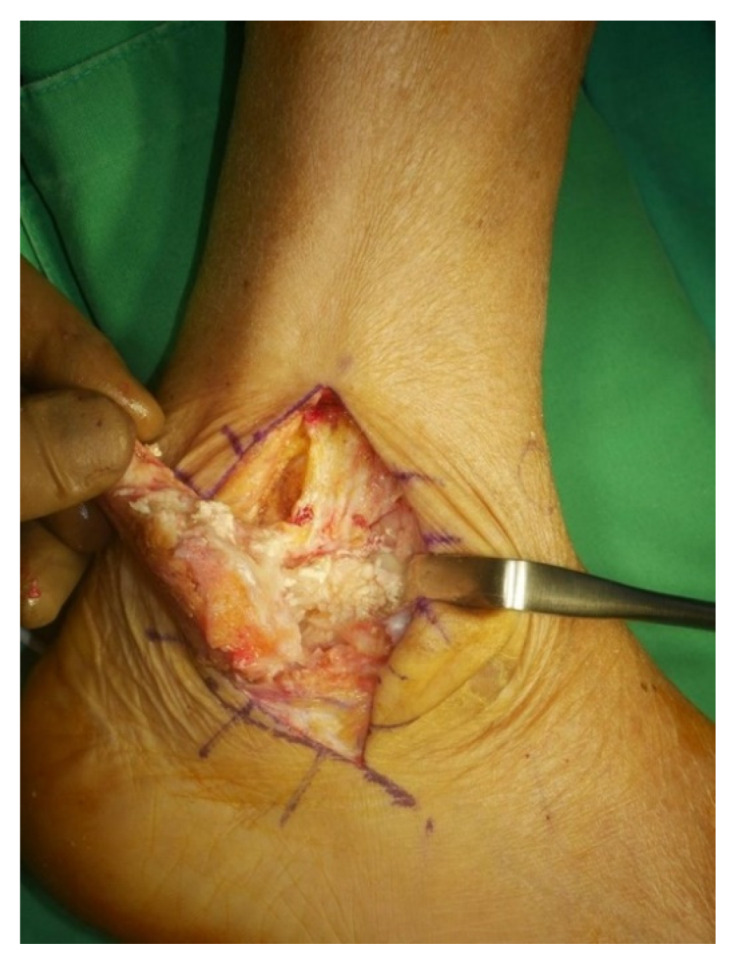
After exposure of the distal fibula, the fibula was transected 4 to 5 cm proximal to the ankle joint.

**Figure 4 jcm-10-05915-f004:**
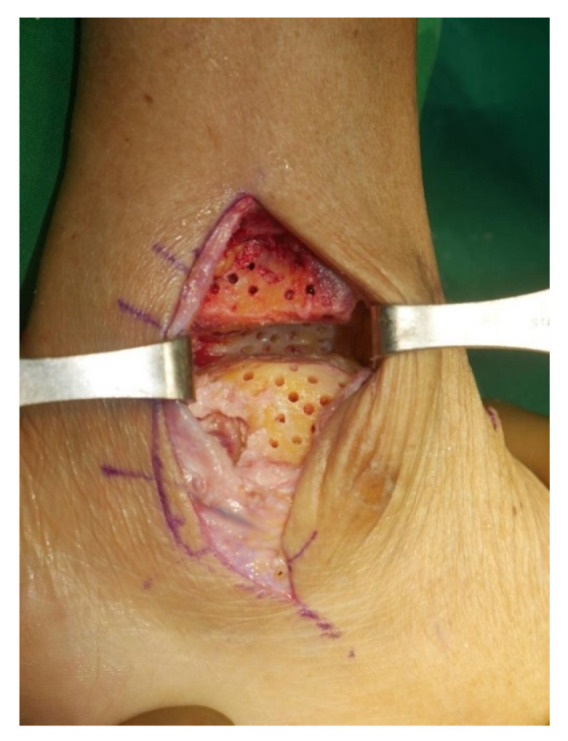
The cartilage and subchondral bone of the tibiotalar joint was removed using curettes, osteotomes, and burrs through the lateral ankle joint.

**Figure 5 jcm-10-05915-f005:**
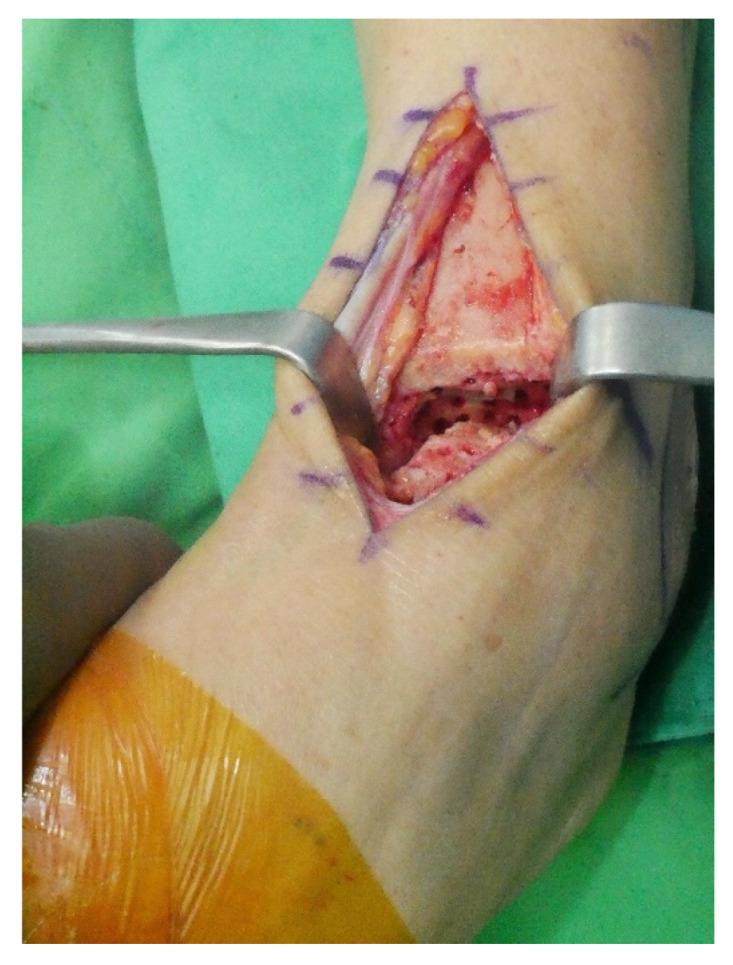
After dissection was performed between the tibialis anterior and extensor halluces longus tendons to expose the joint, cartilage and subchondral bone of the medial gutter were removed.

**Figure 6 jcm-10-05915-f006:**
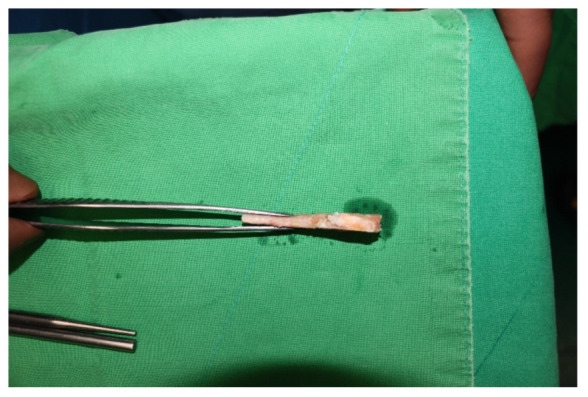
The resected fibular fragment was hemisected in the sagittal plane for use as a bone graft. The medial half was used as the filler of the arthrodesis site, and the lateral half was used as a lateral strut.

**Figure 7 jcm-10-05915-f007:**
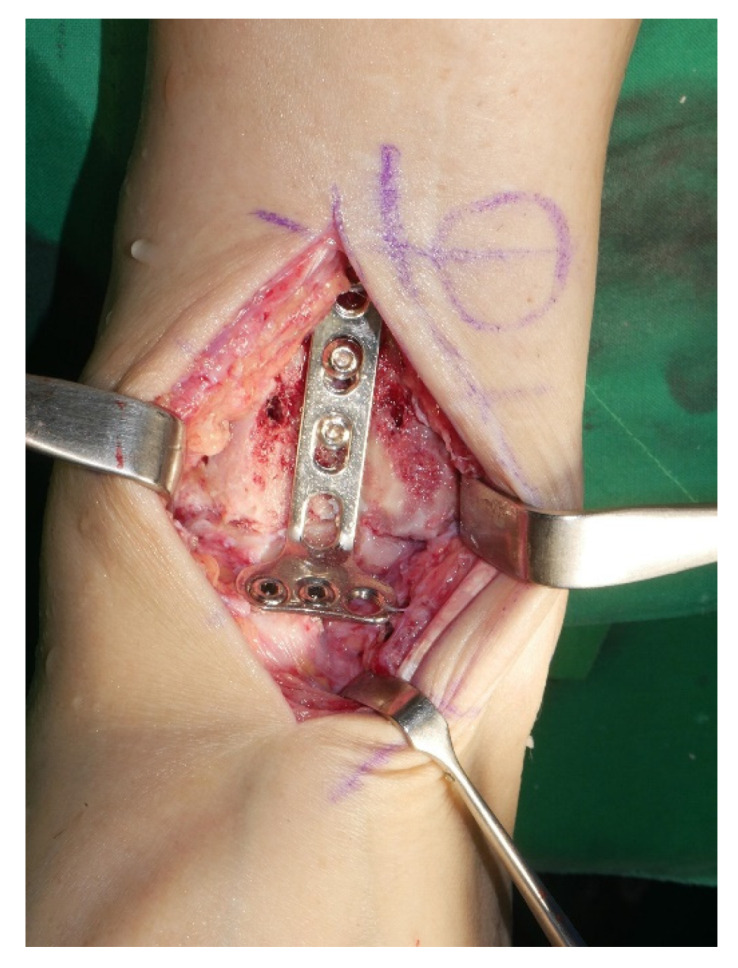
After screw fixation, additional anterior plating was performed through anterior incision.

**Figure 8 jcm-10-05915-f008:**
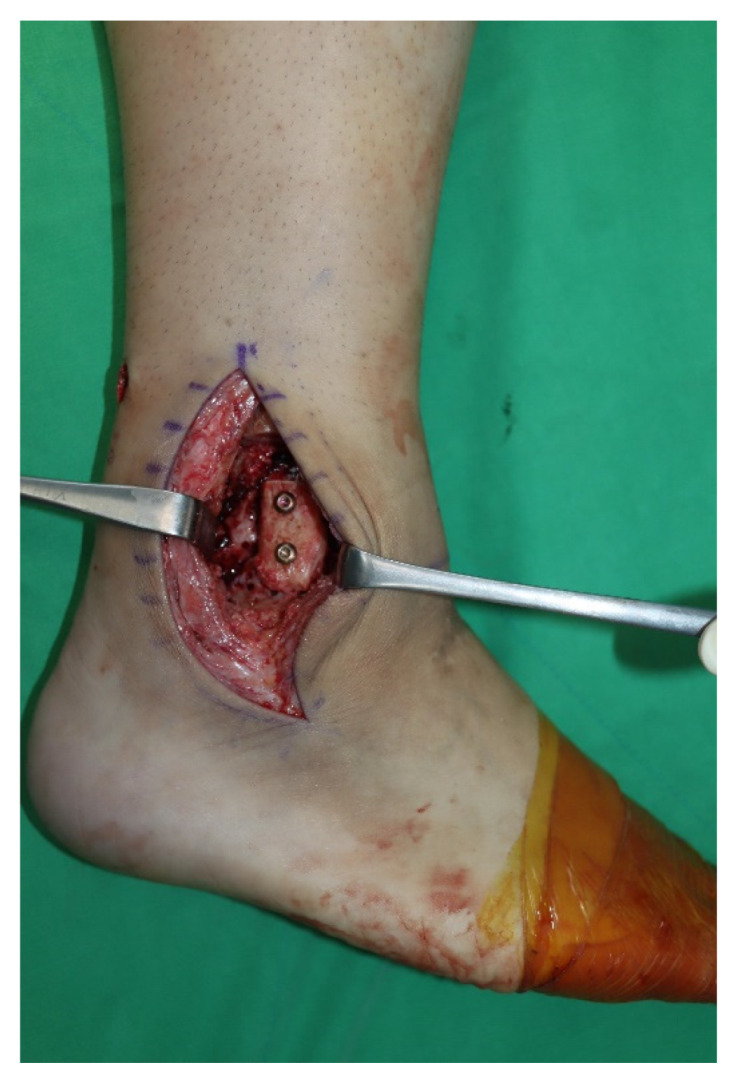
The lateral half of the resected fibula was fixed by four cancellous screws in its original anatomical position as a natural lateral bony plate.

**Figure 9 jcm-10-05915-f009:**
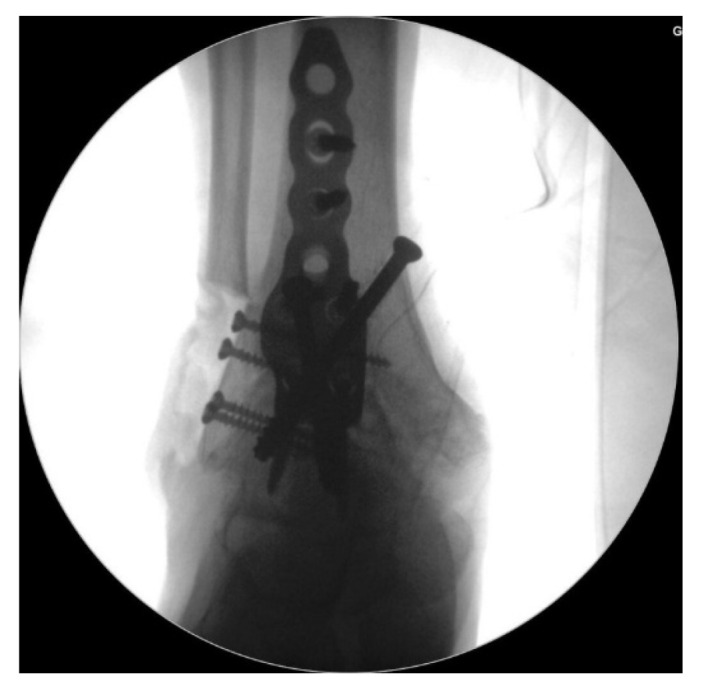
Reduction state and screw positioning without penetration of the subtalar joints were confirmed via fluoroscopy.

**Figure 10 jcm-10-05915-f010:**
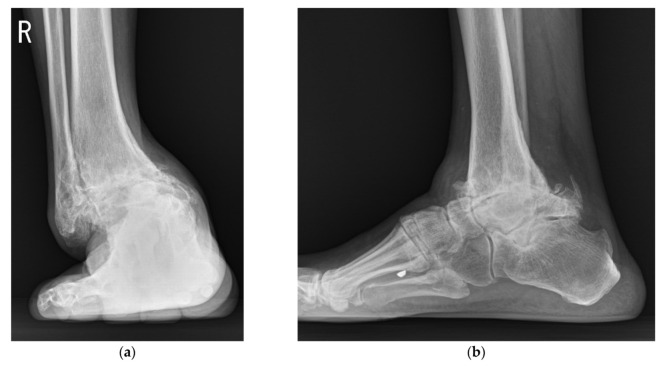
Preoperative weightbearing radiographs showed end-stage ankle arthritis with severe varus deformity: (**a**) anteroposterior view; (**b**) lateral view.

**Figure 11 jcm-10-05915-f011:**
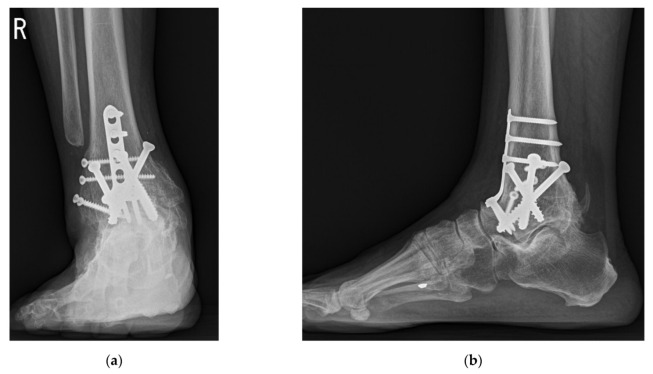
Postoperative weightbearing radiographs taken at 1 year after surgery showed union with improved position: (**a**) anteroposterior view; (**b**) lateral view.

**Table 1 jcm-10-05915-t001:** Pre- and postoperative radiographic results.

Radiographic Parameters	Preoperative	Last Follow-Up	p-Value
Medial distal tibial angle (°)	86.3 ± 6.4 (63~96)		
Talar tilt (°)	5.1 ± 15.1 (−33~33)		
Talus center migration (mm)	3.5 ± 6.3 (18~17)	0.2 ± 1.7 (−4~4)	0.001
Anterior distal tibial angle (°)	74.2 ± 9.4 (54~91)		
Anteroposterior offset (mm)	6.1 ± 5.3 (1~23)		
Sagittal talar migration (mm)	13.6 ± 8.2 (−4~34)	7.9 ± 4.0 (0~16)	<0.001
Hindfoot alignment angle (°)	4.8 ± 15.4 (−36~30)	−1.1 ± 8.8 (−29~13)	0.001

## Data Availability

Data available on request due to restrictions of privacy.

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
