# Peer review of "Combined Transfibular and Anterior Approaches Increase Union Rate and Decrease Non-Weight-Bearing Periods in Ankle Arthrodesis: Combined Approaches in Ankle Arthrodesis"

_jcm, 2021, doi:10.3390/jcm10245915_

Round 1

Reviewer 1 Report

Expand upon complications re anterior impingement and plate bulk; was there a certain proprietary company used in those patients vs others. Were the screw trajectories standardized -- re patients that had post-op screw penetration. Discuss more in introduction re possible advantages/disadvantages of approaches included. Discuss more particular point of correcting coronal plane deformity and functional implications for patients if correction isn't achieved. 

Author Response

Q1. Expand upon complications re anterior impingement and plate bulk; was there a certain proprietary company used in those patients vs others.

A1. As we described in the materials and methods section, a 3.5-mm T-plate (Zimmer Corp, Warsaw, Ind) was used in 29 cases, and anatomical anterior fusion plate (Arthrex Inc., FL, USA) was used in 7 cases.

Q2. Were the screw trajectories standardized -- re patients that had post-op screw penetration. 

A2. We used standard screw fixation technique in ankle arthrodesis. The first screw was placed from the posterior malleolus along the longitudinal axis of the talar neck and into the talar head. A second screw was placed from the posteromedial distal tibia into the anterolateral talar body.

Q3. Discuss more in introduction re possible advantages/disadvantages of approaches included.

A3. We described advantages and disadvantages of several approaches in discussion section as your comments.

Q4. Discuss more particular point of correcting coronal plane deformity and functional implications for patients if correction isn't achieved.

A4. We describe the points you pointed out in more detail in the discussion section.

Reviewer 2 Report

Comments to the authors of the manuscript number jcm-1432237 titled: Combined Transfibular and Anterior Approaches Increase Union Rate and Decrease Non-Weight-Bearing Periods in Ankle Arthrodesis: Combined Approaches in Ankle Arthrodesis.

Authors explain a new surgical procedure in the ankle arthrodesis to decrease the non weight bearing periods and increase the union arthrodesis rate.

This is an interest manuscript well redacted but minor changes are required.

0.- Abstract: Minor changes are required based in the comments of the manuscript.

1.- Introduction

Lines 45-47: Please add reference

Lines 54-55: Please add reference

Lines 55-61: Please add reference

2.- material and methods

- How authors calculate the sample size?

- Lines 74 – 76: Can author explain better exclusion criteria? What about psoriasic arthritis?

- Lines 78 – 80: Can authors explain better? Has diabetes and smoke habits influence in

the arthritis?

- Lines 85 - 113: Can authors add radiographs images?

- Lines 115- 116: Can authors explain based in scientific literature why patients receive antibiotics before surgery? Which are benefits? Are they necessary before surgery? What happens in case of post operative infection?

3.- Results

- Can authors add pre and post radiographs?

- Please add sociodemographic table

- Table 1: Where are the postoperative data results?, please explain what menas (º) in the table.

4.- Discussion: Is well redacted and explained.

5.- Conclusions. Can explain better your conclusions?

Author Response

Thank you very much for your comments and questions on our paper. We are pleased that you are interested in our paper. We hope that we have answered your questions and comments and have made the necessary changes and corrections adequately. We appreciate the thoroughness with which the reviewers regarded our paper and hope that it is now suitable for publication in "Journal of Clinical Medicine".

Reviewer 2

0.- Abstract: Minor changes are required based in the comments of the manuscript.

1.- Introduction

Q1. Lines 45-47: Please add reference

A1. We added reference.

Q2. Lines 54-55: Please add reference

A2. We added reference.

Q3. Lines 55-61: Please add reference

A3. This part is not the content of the previously reported literature, but the subjective description of the authors' experiences. So it is difficult to add reference.

2.- material and methods

Q4. How authors calculate the sample size?

A4. As a case series, sample size was not evaluated in this study. As we described in discussion section, small sample size is a limitation of this study. Therefore, we’re preparing a prospective randomized controlled trial with a large sample size comparing with other technique is needed to minimize the bias.

Q5. Lines 74 – 76: Can author explain better exclusion criteria? What about psoriasic arthritis?

A5. Although the authors did not experience psoriatic arthritis in this study, psoriatic arthritis was included in the exclusion criteria. Psoriatic arthritis was described as an exclusion criterion in the manuscript.

Q6. Lines 78 – 80: Can authors explain better? Has diabetes and smoke habits influence in the arthritis?

A6. It is difficult to explain whether diabetes and smoking influenced the development of arthritis. We describe these as diabetes and tobacco are known risk factors for postoperative wound problems.

Q7. Lines 85 - 113: Can authors add radiographs images?

A7. We added example image for radiographic parameters as your recommendation.

Q8. Lines 115- 116: Can authors explain based in scientific literature why patients receive antibiotics before surgery? Which are benefits? Are they necessary before surgery? What happens in case of post operative infection?

A8. As we know, there are many controversies about the effectiveness of prophylactic antibiotics used before surgery, and there is no established recommendation for foot and ankle surgery. However, we have empirically used prophylactic antibiotics in all surgical patients. Fortunately, no postoperative infection occurred in this study, and no postoperative infection was experienced after all arthrodesis excluded from this study.

3.- Results

Q9. Can authors add pre and post radiographs?

A9. As your comment, we added pre- & postoperative radiographs.

Q10. Please add sociodemographic table

A10. We already added demographic data in materials & methods section. Please let me know if you need any additional information. We’ll add content.

Q11. Table 1: Where are the postoperative data results?, please explain what menas (º) in the table.

A11. As we described in Table 1, the results of the last follow-up refer to the postoperative results. MDTA, talar tilt, ADTA, and anteroposterior offset were not described in the Table 1 because these were impossible to measure in the postoperative radiographs. And we did not understand “explain what means (º) in the table”. Please if you explain this in more detail, I will reply back or add content.

Q12. Discussion: Is well redacted and explained.

A12. Thank you for your compliments.

Q13. Conclusions. Can explain better your conclusions?

A13. We modified conclusions in manuscript as your comment.
